# Spatiotemporal Dynamic Graph Isomorphism Network For Satellite Image Time Series Classification

Yuchen Jin
*School of Control Science and Engineering*
*Dalian University of Technology*
Dalian, China
jinyuchen@mail.dlut.edu.cn

Degang Wang
*School of Control Science and Engineering*
*Dalian University of Technology*
Dalian, China
wangdg@dlut.edu.cn

*Abstract*—**The evolution patterns of land cover types exhibit similarities, which increases the difficulty of high-precision temporal land cover classification tasks. Utilizing satellite image time series (SITS) data can effectively capture the spatiotemporal features of land surface changes. This paper proposes a novel spatiotemporal dynamic graph isomorphism network (STDGIN) for the classification of SITS data. The STDGIN consists of the temporal graph network module and the temporal continuity module. First, each band of SITS is combined into nodes using aggregation convolution, and the dynamic adjacency matrix is designed to enable information transmission between graphs. Then, the hierarchical information of the graph and the local spatiotemporal features of SITS are captured using the node aggregation mechanism. Concurrently, the Bi-LSTM-based module is constructed to capture the temporal continuity features of SITS. Finally, the two features are fused into spatiotemporal representations, and land categories are predicted using a linear classifier. Simulation results indicate that the STDGIN can achieve superior performance on a public SITS dataset (TiSeLaC).**

*Keywords—satellite image time series classification, temporal graph, long short-term memory network, representation learning*

## I. Introduction

With the ongoing changes in the environment, the continual updating of land cover data is crucial for land decision-making. Scientifically classifying land cover types not only improves the quality of surveys and maps, but also helps to organize land use and production. Traditional single-time-point remote sensing data has limitations in distinguishing land cover types with similar structures and spectral features. Therefore, satellite image time series (SITS) data is used to improve the accuracy of land cover classification problems.

Over the past few years, various methods have been used to solve SITS classification problems. Among them, methods utilizing deep learning have achieved significant results in improving SITS classification performance. Some scholars use recurrent neural network-based methods, such as Long Short-Term Memory (LSTM) ([1]) and Bidirectional Long Short-Term Memory (Bi-LSTM) ([2]), to extract temporal continuity features from SITS. Since convolutional neural networks (CNN) perform well on multivariate time series (MTS) data, TempCNN ([3]) is proposed to extract local temporal features with different convolutional kernel sizes. As the attention mechanism has been widely applied in deep learning, the GL-TAE ([4]) utilizes global and local attention encoders to classify SITS data. Besides, some graph neural network-based models are also applied in extracting correlative features from MTS data. Todynet ([5]) is constructed to capture local spatiotemporal dependencies by dynamically constructing graph structures. FC-STGNN ([6]) is designed to capture sensor correlations at different timestamps by using fully connected graph convolution.

Although existing methods achieve impressive classification results, extracting temporal continuity features and local spatiotemporal correlation features from SITS data for classification is still an interesting problem. Driven by these insights, in this paper a novel Spatial-Temporal Dynamic Graph Isomorphism Network (STDGIN) is designed for SITS data classification task. The key contributions of this paper are listed below:

- The proposed STDGIN model performs pixel-level classification of SITS data and solves the issue of classifying multi-temporal Landsat-8 satellite remote sensing images.

- In STDGIN, the aggregation convolution layer combines different bands of SITS data into nodes. And the dynamic weight matrix allows information to flow between different temporal graphs. These information can improve the performance of extracting local spatiotemporal correlation features from SITS.

The rest of this paper is structured as follows: A spatial-temporal graph isomorphism network that jointly captures temporal continuity features and local spatiotemporal correlation features is established in Section II. The effectiveness of the STDGIN is validated through some experiments in Section III. The conclusion is provided in Section IV.

## II. Spatiotemporal Dynamic Graph Isomorphism Network

In this section, the STDGIN model is proposed, and the architecture is shown in Fig. 1. A series of remote sensing

images are the input values, and pixel-level segmentation is used to generate SITS data $\boldsymbol{X} = [\boldsymbol{x}_1, \boldsymbol{x}_2, \ldots, \boldsymbol{x}_L]^\mathrm{T} \in \mathbb{R}^{L \times d}$, where $L$ is the number of time steps, $\boldsymbol{x}_l = [x_{l,1}, x_{l,2}, \ldots, x_{l,d}]^\mathrm{T} \in \mathbb{R}^{1 \times d}$ is the band vector at time step $l (l = 1,2, \ldots, L)$, and $d$ is the number of band sensors at each time step. The STDGIN is split into two components: the temporal graph module and the temporal continuity module. The input $\boldsymbol{X}$ is processed by the temporal graph module to obtain the local spatiotemporal correlation features vector $\boldsymbol{S}_G$, and by the temporal continuity module to obtain the temporal continuity features vector $\boldsymbol{Z}_G$. Then, these features are concatenated to construct the spatiotemporal representation vector $\boldsymbol{U}$. Finally, $\boldsymbol{U}$ is passed through a linear layer to obtain the model's prediction output $\hat{\boldsymbol{y}}$.

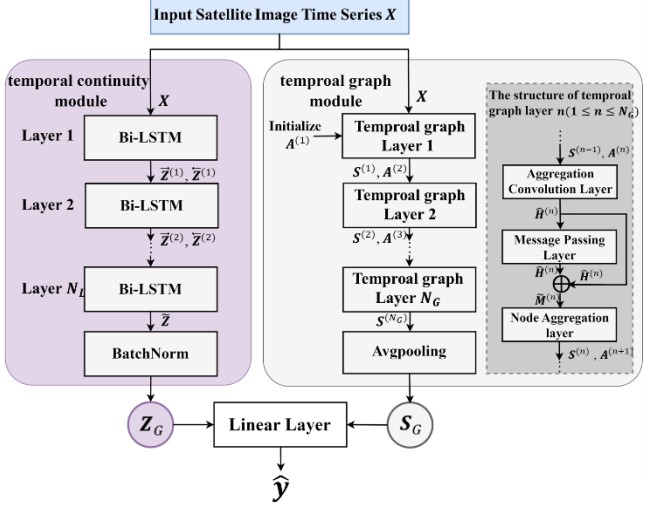

Fig. 1.   Architecture of the STDGIN model

### A. Temporal Graph Module

The temporal graph module is composed of $N_G$ stacked aggregation convolutional layers, message passing layers, and node aggregation layers.

Building on the principles of 1D-CNN ([7]), which effectively captures temporal features through convolutional operations, the sliding convolutional kernel in the aggregation convolutional layer is employed to extract local spatiotemporal correlation features from SITS data. The input SITS data $\boldsymbol{X}$ is equally divided into $T$ time slots by $s$, denoted by $\widetilde{\boldsymbol{X}} = [\widetilde{\boldsymbol{x}}_1, \widetilde{\boldsymbol{x}}_2, \ldots, \widetilde{\boldsymbol{x}}_T]^\mathrm{T} \in \mathbb{R}^{T \times s \times d}$, where $s$ is a factor of the time steps $L$, and $\widetilde{\boldsymbol{x}}_t = [\boldsymbol{x}_{(t-1)s+1}, \ldots, \boldsymbol{x}_{ts}]^\mathrm{T} \in \mathbb{R}^{s \times d}$. Since the structure of the initial layer in the aggregation convolutional layer is different from the subsequent layers, the aggregation convolution operations of the initial layer and the subsequent layers are explained separately.

The generation process of the node feature $\boldsymbol{h}_{t,i}^{(1)} \in \mathbb{R}^{1 \times d^{(1)}}$ for the $i$-th $(i = 1,2, \ldots, C^{(1)})$ channel in the initial layer is expressed as follows:

$$\boldsymbol{h}_{t,i}^{(1)} = \widetilde{\boldsymbol{x}}_t * \boldsymbol{W}_{H,i}^{(1)} + \boldsymbol{b}_{H,i}^{(1)}$$
$$= \sum_{j=1}^{s} \boldsymbol{w}_{H,i,j}^{(1)} \odot \boldsymbol{x}_{(t-1) \cdot s + j} + \boldsymbol{b}_{H,i}^{(1)} \tag{1}$$

where $C^{(1)}$ is the number of channels expanded by the aggregation convolution in the initial layer, $\boldsymbol{W}_{H,i}^{(1)} = [\boldsymbol{w}_{H,i,1}^{(1)}, \boldsymbol{w}_{H,i,2}^{(1)}, \ldots, \boldsymbol{w}_{H,i,s}^{(1)}]^\mathrm{T} \in \mathbb{R}^{s \times d}$ is the convolutional kernel of the $i$-th channel, $\boldsymbol{w}_{H,i,j}^{(1)} \in \mathbb{R}^{1 \times d}$ is the element at the $j$-th position in the convolutional kernel, $\odot$ is the Hadamard product, $\boldsymbol{b}_{H,i}^{(1)} \in \mathbb{R}^{1 \times d}$ is the bias in the convolutional kernel of the $i$-th channel, and the output of all channels in the initial layer is $\boldsymbol{H}_t^{(1)} = [\boldsymbol{h}_{t,1}^{(1)}, \boldsymbol{h}_{t,2}^{(1)}, \ldots, \boldsymbol{h}_{t,C^{(1)}}^{(1)}]^\mathrm{T} \in \mathbb{R}^{C^{(1)} \times d}$.

The generation process of the node feature $\boldsymbol{h}_{t,i}^{(n)} \in \mathbb{R}^{1 \times d^{(n)}}$ for the $i$-th $(i = 1,2, \ldots, C^{(n)})$ channel in the $n$-th $(n = 2,3, \ldots, N_G)$ layer is expressed as:

$$\boldsymbol{h}_{t,i}^{(n)} = \boldsymbol{S}_t^{(n-1)^\mathrm{T}} * \boldsymbol{W}_H^{(n)} + \boldsymbol{b}_H^{(n)}$$
$$= \sum_{i=0}^{C^{(n)}} \left( \boldsymbol{W}_{H,i}^{(n)} \odot \boldsymbol{S}_t^{(n-1)^\mathrm{T}} + \boldsymbol{b}_{H,i}^{(n)} \right) \tag{2}$$

where $\boldsymbol{S}_t^{(n-1)} = [\boldsymbol{S}_{t,1}^{(n-1)}, \boldsymbol{S}_{t,2}^{(n-1)}, \ldots, \boldsymbol{S}_{t,d^{(n)}}^{(n-1)}]^\mathrm{T} \in \mathbb{R}^{d^{(n)} \times C^{(n-1)}}$ is the output of the previous layer at time slot $t$, $\boldsymbol{S}_t^{(0)} = \widetilde{\boldsymbol{x}}_t^\mathrm{T}$ and $\boldsymbol{W}_H^{(n)} = [\boldsymbol{W}_{H,1}^{(n)}, \boldsymbol{W}_{H,2}^{(n)}, \ldots, \boldsymbol{W}_{H,C^{(n)}}^{(n)}]^\mathrm{T} \in \mathbb{R}^{C^{(n)} \times C^{(n-1)} \times d^{(n)}}$ is the convolutional kernel of the $n$-th layer. $\boldsymbol{W}_{H,i}^{(n)} \in \mathbb{R}^{C^{(n-1)} \times d^{(n)}}$ is the element of $\boldsymbol{W}_H^{(n)}$ for the $i$-th channel. The output of all channels in the $n$-th layer is $\boldsymbol{H}_t^{(n)} = [\boldsymbol{h}_{t,1}^{(n)}, \boldsymbol{h}_{t,2}^{(n)}, \ldots, \boldsymbol{h}_{t,C^{(n)}}^{(n)}]^\mathrm{T} \in \mathbb{R}^{C^{(n)} \times d^{(n)}}$.

At this stage, it is considered that $T$ is the count of graphs in the temporal graph, $C^{(n)}$ is the feature dimension of each graph node in the $n$-th layer, and $d^{(n)}(d^{(1)} = d)$ is the count of nodes in the $n$-th layer. Let $\widehat{\boldsymbol{H}}_t^{(n)}$ be the transpose of $\boldsymbol{H}_t^{(n)^\mathrm{T}}$. The output of the aggregation convolutional layer is $\widehat{\boldsymbol{H}}^{(n)} = [\widehat{\boldsymbol{H}}_1^{(n)}, \widehat{\boldsymbol{H}}_2^{(n)}, \ldots, \widehat{\boldsymbol{H}}_T^{(n)}]^\mathrm{T}$, where $\widehat{\boldsymbol{H}}_t^{(n)} = [\widehat{\boldsymbol{h}}_{t,1}^{(n)}, \widehat{\boldsymbol{h}}_{t,2}^{(n)}, \ldots, \widehat{\boldsymbol{h}}_{t,d^{(1)}}^{(n)}]^\mathrm{T}$, and $\widehat{\boldsymbol{h}}_{t,v}^{(n)}$ is the node feature of the $v$-th $(v = 1,2, \ldots, d^{(n)})$ node at time slot $t$ in the $n$-th layer.

Since there is no predefined adjacency matrix for SITS data, the initial adjacency matrix $\boldsymbol{A}^{(1)} = [\boldsymbol{A}_1^{(1)}, \boldsymbol{A}_2^{(1)}, \ldots, \boldsymbol{A}_T^{(1)}]^\mathrm{T} \in \mathbb{R}^{T \times d^{(1)} \times d^{(1)}}$ is constructed using the random initialization method ([5]). Specifically, $\boldsymbol{A}_t^{(1)}$ is the adjacency matrix for the time slot $t(t = 1,2, \ldots, T)$ in the initial layer. To ensure that nodes retain their own features during each message-passing process and do not rely entirely on the features of neighboring nodes, the adjacency matrix $\boldsymbol{A}^{(n)}$ is symmetrically normalized to generate $\widehat{\boldsymbol{A}}_t^{(n)} \in \mathbb{R}^{d^{(n)} \times d^{(n)}}$.

$$\widehat{\boldsymbol{A}}_t^{(n)} = \widetilde{\boldsymbol{D}}_t^{(n)^{-\frac{1}{2}}} (\boldsymbol{A}_t^{(n)} + \boldsymbol{I}) \widetilde{\boldsymbol{D}}_t^{(n)^{-\frac{1}{2}}} \tag{3}$$

where $\widetilde{\boldsymbol{D}}_t^{(n)}$ represents the degree matrix of $(\boldsymbol{A}_t^{(n)} + \boldsymbol{I})$, and $\boldsymbol{I}$ represents the identity matrix, and the normalized adjacency matrix is $\widehat{\boldsymbol{A}}^{(n)} = [\widehat{\boldsymbol{A}}_1^{(n)}, \widehat{\boldsymbol{A}}_2^{(n)}, \ldots, \widehat{\boldsymbol{A}}_T^{(n)}]^\mathrm{T} \in \mathbb{R}^{T \times d^{(n)} \times d^{(n)}}$.

Then, $\widetilde{\boldsymbol{H}}^{(n)}$ and $\widehat{\boldsymbol{A}}_t^{(n)}$ are input into the message passing layer. Graph isomorphism networks (GIN) ([8]) are used for processing static graph data and perform well in node message passing. To improve their ability to handle SITS data and to be

applicable to temporal graph structures, dynamic adjacency matrices are designed for GIN. This improvement allows the model to better capture the dynamic flow of information between different time slots and enhances its effectiveness in SITS analysis. Specifically, dynamic adjacency matric $\widehat{\boldsymbol{Q}}^{(n)} = [\widehat{\boldsymbol{Q}}_1^{(n)}, \widehat{\boldsymbol{Q}}_2^{(n)}, ..., \widehat{\boldsymbol{Q}}_{T-1}^{(n)}]^{\mathrm{T}} \in \mathbb{R}^{(T-1)\times d^{(n)} \times d^{(n)}}$ is designed for each layer using the Xavier method, where $\widehat{\boldsymbol{Q}}_t^{(n)}$ is the dynamic weight matrix for the $t$-th time slot graph relative to the $(t+1)$-th time slot graph in the $n$-th layer. The message passing equation is expressed as:

$$\widetilde{\boldsymbol{h}}_{t,v}^{(n)} = \left(1+\epsilon^{(n)}\right)\widehat{\boldsymbol{h}}_{t,v}^{(n)} + \sum_{i=0}^{d^{(n)}} \hat{a}_{t,i,v}^{(n)}\,\widehat{\boldsymbol{h}}_{t,v}^{(n)}$$
$$+ \sum_{j=0}^{d^{(n)}} \hat{q}_{t-1,j,v}^{(n)}\,\widehat{\boldsymbol{h}}_{t-1,v}^{(n)} \tag{4}$$

where, $\widetilde{\boldsymbol{h}}_{t,v}^{(n)} \in \mathbb{R}^{1\times C^{(n)}}$, $\epsilon^{(n)}$ is a learnable parameter. $\hat{a}_{t,i,v}^{(n)}$ denotes the element in the $i$-th row and $v$-th column of the adjacency matrix $\widehat{\boldsymbol{A}}_t^{(n)}$. $\hat{q}_{t-1,j,v}^{(n)}$ denotes the element in the $j$-th row and $v$-th column of the adjacency matrix $\widehat{\boldsymbol{Q}}_{t-1}^{(n)}$, and $\widehat{\boldsymbol{h}}_{t-1,v}^{(n)}$ is the feature of the $v$-th node in the $(t-1)$-th time slot. After message passing, all node features are obtained which are denoted by $\widetilde{\boldsymbol{H}}_t^{(n)} = [\widetilde{\boldsymbol{h}}_{t,1}^{(n)}, \widetilde{\boldsymbol{h}}_{t,2}^{(n)}, ..., \widetilde{\boldsymbol{h}}_{t,d^{(n)}}^{(n)}]^{\mathrm{T}}$.

Furthermore, to address the problem of gradient vanishing and exploding, and to improve the model's expressiveness and training stability, residual connections are used between the aggregation convolution layer and the message-passing layer. Residual connections generate the input $\widetilde{\boldsymbol{M}}_t^{(n)} = \widehat{\boldsymbol{H}}_t^{(n)} + \widetilde{\boldsymbol{H}}_t^{(n)}$ for the node aggregation layer.

In the node aggregation layer, $\widetilde{\boldsymbol{M}}_t^{(n)}$ is input into the node aggregation layer, where the DiffPool method ([9]) is considered to aggregate nodes and refine the hierarchical representation of the graph structure. The process of generating the output $\boldsymbol{S}_t^{(n)} \in \mathbb{R}^{d^{(n+1)}\times C^{(n)}}$ for the $t$-th time slot in the $n$-th layer is computed by the following equation:

$$\boldsymbol{S}_t^{(n)} = \phi_{ReLU}(\boldsymbol{P}_t^{(n)}\widetilde{\boldsymbol{M}}^{(n)}) \tag{5}$$

where, $\phi_{ReLU}(\cdot)$ is the ReLU activation function, $\boldsymbol{P}^{(n)} = [\boldsymbol{P}_1^{(n)}, \boldsymbol{P}_2^{(n)}, ..., \boldsymbol{P}_T^{(n)}]^{\mathrm{T}} \in \mathbb{R}^{T\times d^{(n+1)}\times d^{(n)}}$ represents the learnable intermediate matrix for the $n$-th layer, initialized by the Xavier method. The adjacency matrix for the $t$-th time slot in the next layer $\boldsymbol{A}_t^{(n+1)}$ is computed by the following equation:

$$\boldsymbol{A}_t^{(n+1)} = \boldsymbol{P}_t^{(n)}\boldsymbol{A}_t^{(n)}\boldsymbol{P}_t^{(n)^{\mathrm{T}}} \tag{6}$$

$\boldsymbol{S}^{(n)}$ and $\boldsymbol{A}_t^{(n+1)}$ are input into the aggregation convolution layer of the $(n+1)$-th layer, and this process is repeated until the $N_G$-th layer. The feature $\boldsymbol{S}^{(N_G)} = [\boldsymbol{S}_1^{(N_G)}, \boldsymbol{S}_2^{(N_G)}, ..., \boldsymbol{S}_T^{(N_G)}]^{\mathrm{T}} \in \mathbb{R}^{T\times d^{(N_G+1)}\times C^{(N_G)}}$ of the entire graph can be obtained, where $d^{(N_G+1)}$ indicates the count of nodes in the final layer's output and $C^{(N_G)}$ represents the feature dimension of the nodes in the final layer.

In the average pooling layer, $\boldsymbol{S}^{(N_G)}$ is sent to average pooling to obtain the output feature of the graph learning module, $\boldsymbol{S}_G \in \mathbb{R}^{1\times d_G}$. The process of the average pooling operation is shown as follows:

$$\boldsymbol{S}_G = \frac{1}{T}\sum_{t=1}^{T}\boldsymbol{S}_t^{(N_G)} \tag{7}$$

where $T$ represents the number of time slot graphs. Hence, the local spatiotemporal correlation features $\boldsymbol{S}_G$ is obtained.

### B. Temporal Continuity Module

The temporal continuity module is composed of $N_L$ Bi-LSTM layers. The input for the initial layer is the SITS data sample $\boldsymbol{X}$. The output from each layer acts as the input for the subsequent layer. The output of the $N_L$-th layer, after batch normalization, is the temporal continuity features $\boldsymbol{Z}_G$.

The input SITS data is $\boldsymbol{X} = [\boldsymbol{x}_1, \boldsymbol{x}_2, ..., \boldsymbol{x}_L]^{\mathrm{T}} \in \mathbb{R}^{L\times d}$, where $\boldsymbol{x}_l \in \mathbb{R}^{1\times d}$ is the input feature vector at the $l$-th time step $l (l = 1,2,...,L)$. The forward hidden state output of the Bi-LSTM at the $l$-th time step of the $n$-th layer $n (n = 1,2,...,N_L)$ is $\overrightarrow{\boldsymbol{Z}}_l^{(n)} \in \mathbb{R}^{1\times u^{(n)}}$, which is derived from the hidden state output of the previous layer $\overrightarrow{\boldsymbol{Z}}_l^{(n-1)}$ at the previous time step. The feature dimension of the hidden state output at the $n$-th layer is $u^{(n)}$. The initial state $\overrightarrow{\boldsymbol{Z}}_l^{(0)}$ is set to $\boldsymbol{x}_l \in \mathbb{R}^{1\times u^{(0)}}$ with $u^{(0)} = d$.

The LSTM unit consists of five gates: the forget gate $\overrightarrow{\boldsymbol{f}}_l^{(n)} \in \mathbb{R}^{1\times u^{(n)}}$, the output gate $\overrightarrow{\boldsymbol{o}}_l^{(n)} \in \mathbb{R}^{1\times u^{(n)}}$, the input gate $\overrightarrow{\boldsymbol{\iota}}_l^{(n)} \in \mathbb{R}^{1\times u^{(n)}}$, the candidate cell state $\overrightarrow{\widetilde{\boldsymbol{c}}}_l^{(n)} \in \mathbb{R}^{1\times u^{(n)}}$, and the updated cell state $\boldsymbol{c}_l^{(n)} \in \mathbb{R}^{1\times u^{(n)}}$. The values of these gates are computed from the previous layer's output hidden state and the hidden state of the current layer at the previous time step. These operations can be summarized by the following equations:

$$\overrightarrow{\boldsymbol{f}}_l^{(n)} = \phi_{sigmoid}\left(\overrightarrow{\boldsymbol{Z}}_l^{(n-1)}\overrightarrow{\boldsymbol{W}}_{f,l}^{(n)} + \overrightarrow{\boldsymbol{R}}_{f,l}^{(n)}\overrightarrow{\boldsymbol{Z}}_{l-1}^{(n)} + \overrightarrow{\boldsymbol{b}}_{f,l}^{(n)}\right) \tag{8}$$

$$\overrightarrow{\boldsymbol{o}}_l^{(n)} = \phi_{sigmoid}\left(\overrightarrow{\boldsymbol{Z}}_l^{(n-1)}\overrightarrow{\boldsymbol{W}}_{o,l}^{(n)} + \overrightarrow{\boldsymbol{R}}_{o,l}^{(n)}\overrightarrow{\boldsymbol{Z}}_{l-1}^{(n)} + \overrightarrow{\boldsymbol{b}}_{o,l}^{(n)}\right) \tag{9}$$

$$\overrightarrow{\boldsymbol{\iota}}_l^{(n)} = \phi_{sigmoid}\left(\overrightarrow{\boldsymbol{Z}}_l^{(n-1)}\overrightarrow{\boldsymbol{W}}_{i,l}^{(n)} + \overrightarrow{\boldsymbol{R}}_{i,l}^{(n)}\overrightarrow{\boldsymbol{Z}}_{l-1}^{(n)} + \overrightarrow{\boldsymbol{b}}_{i,l}^{(n)}\right) \tag{10}$$

$$\overrightarrow{\widetilde{\boldsymbol{c}}}_l^{(n)} = \phi_{tanh}\left(\overrightarrow{\boldsymbol{Z}}_l^{(n-1)}\overrightarrow{\boldsymbol{W}}_{\tilde{c},l}^{(n)} + \overrightarrow{\boldsymbol{R}}_{\tilde{c},l}^{(n)}\overrightarrow{\boldsymbol{Z}}_{l-1}^{(n)} + \overrightarrow{\boldsymbol{b}}_{\tilde{c},l}^{(n)}\right) \tag{11}$$

$$\overrightarrow{\boldsymbol{c}}_l^{(n)} = \overrightarrow{\boldsymbol{f}}_l^{(n)}\odot\overrightarrow{\boldsymbol{c}}_{l-1}^{(n)} + \overrightarrow{\boldsymbol{\iota}}_l^{(n)}\odot\overrightarrow{\widetilde{\boldsymbol{c}}}_l^{(n)} \tag{12}$$

$$\overrightarrow{\boldsymbol{Z}}_l^{(n)} = \overrightarrow{\boldsymbol{o}}_l^{(n)}\odot\phi_{tanh}\left(\overrightarrow{\boldsymbol{c}}_l^{(n)}\right) \tag{13}$$

where, $\phi_{sigmoid}(\cdot)$ denotes the sigmoid activation function, $\phi_{tanh}(\cdot)$ denotes the tanh activation function. During the forward process, $\overrightarrow{\boldsymbol{Z}}_0^{(n)}$ and $\overrightarrow{\boldsymbol{c}}_0^{(n)}$ are initialized as zero vectors. The forward process is handled in the chronological order, while the backward process is handled in the reverse chronological order. The formulas are as follows:

$$\overleftarrow{\boldsymbol{f}}_l^{(n)} = \phi_{sigmoid}\left(\overleftarrow{\boldsymbol{Z}}_l^{(n-1)}\overleftarrow{\boldsymbol{W}}_{f,l}^{(n)} + \overleftarrow{\boldsymbol{R}}_{f,l}^{(n)}\overleftarrow{\boldsymbol{Z}}_{l+1}^{(n)} + \overleftarrow{\boldsymbol{b}}_{f,l}^{(n)}\right) \tag{14}$$

$$\overleftarrow{\boldsymbol{o}}_l^{(n)} = \phi_{sigmoid}\left(\overleftarrow{\boldsymbol{Z}}_l^{(n-1)}\overleftarrow{\boldsymbol{W}}_{o,l}^{(n)} + \overleftarrow{\boldsymbol{R}}_{o,l}^{(n)}\overleftarrow{\boldsymbol{Z}}_{l+1}^{(n)} + \overleftarrow{\boldsymbol{b}}_{o,l}^{(n)}\right) \tag{15}$$

$$\overleftarrow{\boldsymbol{\iota}}_l^{(n)} = \phi_{sigmoid}\left(\overleftarrow{\boldsymbol{Z}}_l^{(n-1)}\overleftarrow{\boldsymbol{W}}_{i,l}^{(n)} + \overleftarrow{\boldsymbol{R}}_{i,l}^{(n)}\overleftarrow{\boldsymbol{Z}}_{l+1}^{(n)} + \overleftarrow{\boldsymbol{b}}_{i,l}^{(n)}\right) \tag{16}$$

$$\overleftarrow{\widetilde{\boldsymbol{c}}}_l^{(n)} = \phi_{tanh}\left(\overleftarrow{\boldsymbol{Z}}_l^{(n-1)}\overleftarrow{\boldsymbol{W}}_{\tilde{c},l}^{(n)} + \overleftarrow{\boldsymbol{R}}_{\tilde{c},l}^{(n)}\overleftarrow{\boldsymbol{Z}}_{l+1}^{(n)} + \overleftarrow{\boldsymbol{b}}_{\tilde{c},l}^{(n)}\right) \tag{17}$$

$$\overleftarrow{\boldsymbol{c}}_l^{(n)} = \overleftarrow{\boldsymbol{f}}_l^{(n)} \odot \overleftarrow{\boldsymbol{c}}_{l+1}^{(n)} + \overleftarrow{\boldsymbol{\iota}}_l^{(n)} \odot \overleftarrow{\widetilde{\boldsymbol{c}}}_l^{(n)} \qquad (18)$$

$$\overleftarrow{\boldsymbol{Z}}_l^{(n)} = \overleftarrow{\boldsymbol{o}}_l^{(n)} \odot \phi_{tanh}\left(\overleftarrow{\boldsymbol{c}}_l^{(n)}\right) \qquad (19)$$

where $\overleftarrow{\boldsymbol{Z}}_L^{(n)}$ and $\overleftarrow{\boldsymbol{c}}_L^{(n)}$ are initialized as zero vectors.

This process is repeated until the forward hidden state $\overrightarrow{\boldsymbol{Z}}_L^{(N_L)} \in \mathbb{R}^{1 \times u^{(N_L)}}$ at the $L$-th time step of the $N_L$-th layer and the backward hidden state $\overleftarrow{\boldsymbol{Z}}_1^{(N_L)} \in \mathbb{R}^{1 \times u^{(N_L)}}$ at the first time step of the $N_L$-th layer are obtained. Then, $\overrightarrow{\boldsymbol{Z}}_L^{(N_L)}$ and $\overleftarrow{\boldsymbol{Z}}_1^{(N_L)}$ are concatenated to get $\widetilde{\boldsymbol{Z}} = [\overrightarrow{\boldsymbol{Z}}_L^{(N_L)}, \overleftarrow{\boldsymbol{Z}}_1^{(N_L)}] \in \mathbb{R}^{1 \times d_Z}$, where $d_Z = 2 \cdot u^{(N_L)}$. Finally, $\widetilde{\boldsymbol{Z}}$ is batch normalized to obtain the temporal continuity features $\boldsymbol{Z}_G \in \mathbb{R}^{1 \times d_Z}$.

### C. Classification and Parameter Learning

After obtaining the local spatiotemporal correlation features $\boldsymbol{S}_G \in \mathbb{R}^{1 \times d_G}$ and the temporal continuity features $\boldsymbol{Z}_G \in \mathbb{R}^{1 \times d_Z}$, they are concatenated along the feature dimension to get the final spatiotemporal representation $\boldsymbol{U} = [\boldsymbol{S}_G, \boldsymbol{Z}_G] \in \mathbb{R}^{1 \times d_U}$, where $d_U = d_G + d_Z$. Then, $\boldsymbol{U}$ is passed into a linear layer to get the predicted classification result $\widehat{\boldsymbol{y}} \in \mathbb{R}^{1 \times D}$:

$$\widehat{\boldsymbol{y}} = \boldsymbol{U}\boldsymbol{W}_y + \boldsymbol{b}_y \qquad (20)$$

where $\boldsymbol{W}_y \in \mathbb{R}^{d_U \times D}$ is the weight parameter of the linear layer and $\boldsymbol{b}_y \in \mathbb{R}^{1 \times D}$ is the bias. Thus, the predicted classification result $\widehat{\boldsymbol{y}}$ for one sample is obtained.

Afterward, the multi-class cross-entropy loss is chosen as the loss function for the model. $B$ represents the number of training samples. The set of true classification result labels is $\boldsymbol{Y} = \{\boldsymbol{y}_1, \boldsymbol{y}_2, \ldots, \boldsymbol{y}_B\}$, and the set of predicted classification results is $\widehat{\boldsymbol{Y}} = \{\widehat{\boldsymbol{y}}_1, \widehat{\boldsymbol{y}}_2, \ldots, \widehat{\boldsymbol{y}}_B\}$. The formula for the multi-class cross-entropy loss is expressed by

$$\mathcal{L} = L_{CE} = -\frac{1}{B} \sum_{i=1}^{B} \sum_{j=1}^{D} y_{i,j} \, log(\hat{y}_{i,j}) \qquad (21)$$

where, $D$ is the count of classes, $\boldsymbol{y}_i = [y_{i,1}, y_{i,2}, \ldots, y_{i,D}]$ is the true label vector for the $i$-th sample $i$, and $\widehat{\boldsymbol{y}}_i = [\hat{y}_{i,1}, \hat{y}_{i,2}, \ldots, \hat{y}_{i,D}]$ is the predicted label vector by the STDGIN. $y_{i,j}$ denotes the true probability that the sample belongs to the $j$-th class $(j = 1,2, \ldots, D)$, and $\hat{y}_{i,j}$ represents the predicted probability the sample being in the $j$-th class.

The weights and bias parameters of the temporal graph module, temporal continuity module, and linear layer can be denoted as $\boldsymbol{W}$. The optimal parameters $\boldsymbol{W}^*$ can be achieved by solving the following optimization problem:

$$\boldsymbol{W}^* = \underset{\boldsymbol{W}}{argmin} \, \mathcal{L}(\boldsymbol{W}) \qquad (22)$$

The Adam ([10]) optimizer is used to solve the problem. According to the parameter update rule, the update formula for the parameters $\boldsymbol{W}$ at the $n_d$-th iteration ($n_d = 1,2, \ldots, N_D$) is as follows:

$$\boldsymbol{W}_{n_d} = \boldsymbol{W}_{n_d-1} - \eta \frac{\widehat{\boldsymbol{\mu}}_{n_d}}{\sqrt{\widehat{\boldsymbol{v}}_{n_d}} - \delta} \qquad (23)$$

$$\widehat{\boldsymbol{\mu}}_{n_d} = \left(\beta_\mu \widehat{\boldsymbol{\mu}}_{n_d-1} + (1 - \beta_\mu)\boldsymbol{g}_{n_d}\right)/\left(1 - \beta_\mu^{(n_d)}\right) \qquad (24)$$

$$\widehat{\boldsymbol{v}}_{n_d} = \left(\beta_v \widehat{\boldsymbol{v}}_{n_d-1} + (1 - \beta_v)(\boldsymbol{g}_{n_d})^2\right)/\left(1 - \beta_v^{(n_d)}\right) \qquad (25)$$

where, $\boldsymbol{W}_{n_d-1}$ represents the weight parameters obtained from the $(n_d - 1)$-th iteration, $\eta$ represents the learning rate, and $\delta$ represents a small constant. $\widehat{\boldsymbol{\mu}}_{n_d}$ and $\widehat{\boldsymbol{v}}_{n_d}$ are the first and second moment estimates computed in the $n_d$-th iteration. $\beta_\mu$ and $\beta_v$ are the hyperparameters, and $\boldsymbol{g}_{n_d}$ and $(\boldsymbol{g}_{n_d})^2$ are the gradient value and its square computed in the $n_d$-th iteration, where $\boldsymbol{g}_{n_d} = \nabla_{\boldsymbol{W}} \mathcal{L}(\boldsymbol{W}_{n_d-1})$.

Additionally, to improve convergence and reduce oscillations, a learning rate reduction strategy is used. Specifically, the learning rate $\eta$ is multiplied by a decay factor $\lambda$ if the validation accuracy does not improve after $N_\lambda$ epochs.

The algorithm steps of STDGIN are as follows:

- Step 1. Define the input data $\boldsymbol{X}$ and the predicted labels $\widehat{\boldsymbol{y}}$, and randomly initialize the model parameters $\boldsymbol{W}$.

- Step 2. Input the SITS data into the temporal graph module to generate the local spatiotemporal correlation features $\boldsymbol{S}_G$ based on equations (1) to (7).

- Step 3. Input the SITS data into the temporal continuity module. Generate forward and backward hidden states for each time step according to equations (8) to (19). Concatenate and normalize these states to obtain the temporal continuity feature $\boldsymbol{Z}_G$

- Step 4. Concatenate the features $\boldsymbol{S}_G$ and $\boldsymbol{Z}_G$. Pass the concatenated features through a fully connected layer to produce the predicted labels $\widehat{\boldsymbol{y}}$.

- Step 5. Compute the cross-entropy loss $\mathcal{L}$ according to equation (21). Optimize the parameters using equations (23) to (25).

## III. EXPERIMENTS

To assess the effectiveness of the STDGIN model in SITS classification tasks, we perform numerical experiments using the publicly available TiSeLaC dataset ([11]).

### A. Dataset Introduction and Experimental Setup

TABLE I. NUMBER OF CROPS IN TiSeLaC DATASET

| ID | CropType | samples |
|---|---|---|
| 1 | Urban Areas | 20000 |
| 2 | Other Built-up Surfaces | 3883 |
| 3 | Forests | 20000 |
| 4 | Sparse Vegetation | 19398 |
| 5 | Rocks and Bare Soil | 15530 |
| 6 | Grassland | 6817 |
| 7 | Sugarcane Crops | 9187 |
| 8 | Other Crops | 1754 |
| 9 | Water | 3118 |
| | total | 99687 |

The TiSeLaC dataset is collected from 23 consecutive Landsat satellite images taken on Réunion Island in 2014. The images have a spatial resolution of 30 meters, and each pixel contains 10 features, including the first 7 bands of Landsat-8 and 3 auxiliary remote sensing indices. The dataset includes 99,687 pixels time series. The dataset is divided randomly with 60% allocated for training, 20% for validation, and 20% for testing, with standardization performed before training. Detailed information about the TiSeLaC dataset in Tab. I.

The evaluation metrics used in the experiments are Overall Accuracy (OA) and the mean Intersection over Union (mIoU) for each class. The model is built using PyTorch 1.9 and trained on an NVIDIA GeForce RTX 3080Ti GPU. The main parameter of the experiments are shown in Tab. II.

TABLE II.        MAIN PARAMETER CONFIGURATIONS OF THE EXPERIMENTS

| Parameter | Value | Description |
|---|---|---|
| $N_E$ | 2000 | Total training epochs |
| $B$ | 128 | The size of each batch |
| $N_G$ | 5 | Number of layers in temporal graph |
| $N_L$ | 4 | Number of layers in temporal continuity |
| $\eta_a$ | 0.0001 | Initial learning rate of the optimizer |
| $\lambda$ | 0.5 | Learning rate decay parameter |
| $N_\lambda$ | 50 | Learning rate decay epochs |
| $d^{(1)},\ldots,d^{(N_G)}$ | $\{d, 8, 6, 4, 2\}$ | Number of nodes in each layer |
| $s$ | 3 | Factor for the number of time step |

### B. Comparison with Other Models

To validate our model's effectiveness, we compare it with GL-TAE ([4]), Todynet ([5]), and FC-STGNN ([6]). All models are evaluated using a five-fold cross-validation scheme. In each validation, all models are tested with configurations that achieve the highest accuracy on the validation set. The GL-TAE model consists of 5 layers with feedforward dimensions {128,256,512,1024,2048}, 4 attention heads, and kernel size is 3. The results of GL-TAE are chosen based on the results from reference ([4]).The Todynet model consists of 4 graphs, with kernel size of {9,5,3}, and a pooling rate of 0.2. The FC-STGNN model consists of a convolution kernel size of 6 and a decay rate of 0.7 for the decay matrix. Todynet and FC-STGNN are set with 2000 epochs and a batch size of 128, while other parameters, such as the number of model layers and nodes, are set according to the recommendations in the references ([5-6]).

TABLE III.        EXPERIMENTAL RESULTS ON THE TISELAC DATASETS

| Class | GL-TAE | TodyNet | FC-STGNN | Ours |
|---|---|---|---|---|
| 1 | 93.12±0.38 | 94.28±0.58 | 92.63±0.58 | **94.71±0.48** |
| 2 | 79.97±1.17 | 79.50±1.17 | 74.93±1.86 | **80.96±0.90** |
| 3 | 93.46±0.99 | **93.73±0.14** | 92.61±0.25 | 93.56±0.15 |
| 4 | 96.14±0.47 | 96.44±0.47 | 95.43±0.52 | **96.93±0.45** |
| 5 | 96.73±0.27 | **98.20±0.46** | 97.33±0.17 | 98.07±0.21 |
| 6 | 90.86±1.58 | 93.33±1.12 | 90.66±0.38 | **94.85±0.33** |
| 7 | 95.90±0.34 | 96.50±0.48 | 94.49±0.29 | **96.65±0.21** |
| 8 | **81.23±2.34** | 74.30±4.07 | 66.67±1.70 | 77.58±3.23 |
| 9 | 92.05±1.89 | 92.11±0.93 | 88.98±1.42 | **93.11±0.62** |
| OA(%) | 93.70±0.17 | 94.32±0.18 | 92.63±0.22 | **94.71±0.19** |
| mIoU(%) | 83.42±0.42 | 85.03±0.43 | 80.88±0.16 | **86.16±0.21** |

The experimental results are presented in Tab. III. It is observed that the proposed method achieves satisfactory performance on the overall accuracy and the mIoU index. It also can achieve high classification accuracy.

### C. Ablation Experiments

To assess the contribution of the components in STDGIN, ablation experiments are conducted. The simulation results are shown in Tab. IV.

TABLE IV.        RESULT OF ABLATION EXPERIMENTS

| Ablation modules | OA(%) | mIoU(%) | Rank |
|---|---|---|---|
| Ours | **94.97** | **86.29** | 1 |
| w/o temporal continuity module | 94.50 | 85.05 | 3 |
| w/o Node aggregation | 91.03 | 76.65 | 5 |
| w/o Dynamic adjacency matrix | 94.53 | 85.25 | 2 |
| w/o temporal graph structure | 93.01 | 81.93 | 4 |

The simulation results show that using the temporal graph structure significantly can improve classification performance by capturing hidden dependencies between bands. The node aggregation layer enhances the feature representation capability of the model.

### D. Hyperparameter Experiment

To verify the model's stability under different parameter settings, four key parameters are selected for experiments on the TiSeLaC dataset. The experimental results are shown in Figs. 2-5.

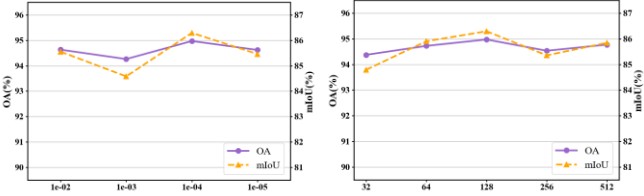

Fig. 2. Results under different $\eta_a$          Fig. 3. Results under different $B$

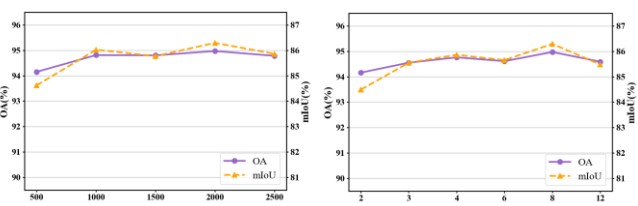

Fig. 4. Results under different $N_E$          Fig. 5. Results under different $T$

It can be observed that within a certain range of hyperparameters, the STDGIN can achieve stable classification accuracy.

Specifically, $N_G$ is set to {2,3,4,5}, and the corresponding model parameters are {51k, 107k, 166k, 225k}. By setting different numbers of layers, the complexity of the model increases accordingly, and the experimental results are shown in Fig. 6. Due to the increase of layers $N_G$, more complex patterns can be captured by the proposed model, thereby enhancing the accuracy of classification or prediction.

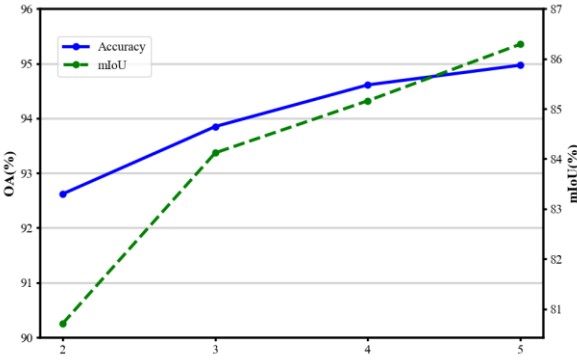

Fig. 6. Results under different $N_E$

*E. Inspection of Class Prototype*

Further, the two-dimensional visualization images are provided in Fig. 7(a) and Fig. 7(b) by the T-SNE algorithm ([12]). It can be seen that the proposed model make the distances between same samples closer.

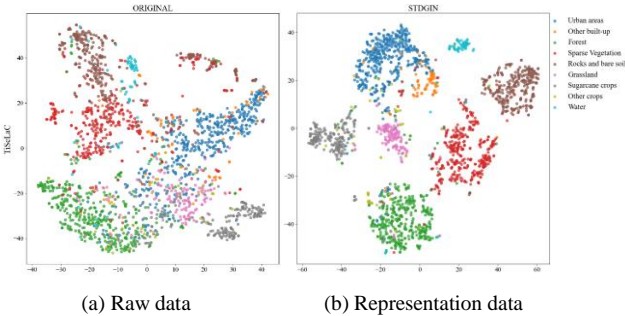

(a) Raw data          (b) Representation data

Fig. 7. T-SNE visualization of the TiSeLaC dataset

## IV. CONCLUSIONS

In this paper, a classification model for SITS data is proposed that combines local spatiotemporal correlation features with temporal continuity features. This approach improves the accuracy of SITS classification. Future research will focus on methods for classifying few-labeled and unlabeled SITS data.

## ACKNOWLEDGMENT

This work was supported by the National Key Research and Development Program of China (2018AAA0100300) and the National Natural Science Foundation (NNSF) of China (12071056).

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
