# OpenReview forum: "Spatiotemporal Dynamic Graph Isomorphism Network For Satellite Image Time Series Classification"
_IEEE.org/ICIST/2024/Conference — IEEE ICIST 2024 Conference Submission_

### Official Review · Reviewer_7Tmr · 2024-08-21
**Spatiotemporal Dynamic Graph Isomorphism Network For Satellite Image Time Series Classification**

**Rating:** 7
**Confidence:** 2

**Review:**

This paper proposes a classification model for SITS data that combines local spatiotemporal correlation features with temporal continuity features. The proposed method improves the accuracy of SITS classification. The research method used is correct. The specific design process is clearly provided. However, it would be better to include quantitative comparisons in the simulation results. Moreover, the computational complexity of the proposed approach should be discussed.

---

### Official Review · Reviewer_fA4a · 2024-08-21
**Accept**

**Rating:** 7
**Confidence:** 5

**Review:**

This paper investigates the use of Spatiotemporal Dynamic Graph Isomorphism Network (STDGIN) for classifying satellite image time series (SITS) data. By using aggregation convolution to combine each band of SITS into nodes and designing a dynamic adjacency matrix for information transmission between graphs, the method captures hierarchical information of the graph and local spatiotemporal features of SITS through a node aggregation mechanism. The paper is well-organized, highly readable, and logically clear. It is recommended to include relevant graphs in the experimental section to clearly highlight the experimental results, and to pay attention to writing conventions.

---

### Official Review · Reviewer_cr7G · 2024-08-22
**Accept**

**Rating:** 7
**Confidence:** 5

**Review:**

This paper proposes a novel spatiotemporal dynamic graph isomorphism network (STDGIN) for the classification of SITS data which improves the accuracy of SITS classification. The paper is logical and reasonable, and the comparison results with several papers are shown in the simulation section.

---

### Decision · Program_Chairs · 2024-09-08

Accept (Oral)